# GRAPH ATTENTION NETWORKS

**Petar Veličković**[*]
Department of Computer Science and Technology
University of Cambridge
`petar.velickovic@cst.cam.ac.uk`

**Guillem Cucurull**[*]
Centre de Visió per Computador, UAB
`gcucurull@gmail.com`

**Arantxa Casanova**[*]
Centre de Visió per Computador, UAB
`ar.casanova.8@gmail.com`

**Adriana Romero**
Montréal Institute for Learning Algorithms
Facebook AI Research
`adrianars@fb.com`

**Pietro Liò**
Department of Computer Science and Technology
University of Cambridge
`pietro.lio@cst.cam.ac.uk`

**Yoshua Bengio**
Montréal Institute for Learning Algorithms
`yoshua.umontreal@gmail.com`

## ABSTRACT

We present graph attention networks (GATs), novel neural network architectures that operate on graph-structured data, leveraging masked self-attentional layers to address the shortcomings of prior methods based on graph convolutions or their approximations. By stacking layers in which nodes are able to attend over their neighborhoods' features, we enable (implicitly) specifying different weights to different nodes in a neighborhood, without requiring any kind of computationally intensive matrix operation (such as inversion) or depending on knowing the graph structure upfront. In this way, we address several key challenges of spectral-based graph neural networks simultaneously, and make our model readily applicable to inductive as well as transductive problems. Our GAT models have achieved or matched state-of-the-art results across four established transductive and inductive graph benchmarks: the *Cora*, *Citeseer* and *Pubmed* citation network datasets, as well as a *protein-protein interaction* dataset (wherein test graphs remain unseen during training).

## 1 INTRODUCTION

Convolutional Neural Networks (CNNs) have been successfully applied to tackle problems such as image classification (He et al., 2016), semantic segmentation (Jégou et al., 2017) or machine translation (Gehring et al., 2016), where the underlying data representation has a grid-like structure. These architectures efficiently reuse their local filters, with learnable parameters, by applying them to all the input positions.

However, many interesting tasks involve data that can not be represented in a grid-like structure and that instead lies in an irregular domain. This is the case of 3D meshes, social networks, telecommunication networks, biological networks or brain connectomes. Such data can usually be represented in the form of graphs.

There have been several attempts in the literature to extend neural networks to deal with arbitrarily structured graphs. Early work used recursive neural networks to process data represented in graph domains as directed acyclic graphs (Frasconi et al., 1998; Sperduti & Starita, 1997). Graph Neural Networks (GNNs) were introduced in Gori et al. (2005) and Scarselli et al. (2009) as a generalization of recursive neural networks that can directly deal with a more general class of graphs, e.g. cyclic, directed and undirected graphs. GNNs consist of an iterative process, which propagates the node

---

[*]Work performed while the author was at the Montréal Institute of Learning Algorithms.

states until equilibrium; followed by a neural network, which produces an output for each node based on its state. This idea was adopted and improved by Li et al. (2016), which propose to use gated recurrent units (Cho et al., 2014) in the propagation step.

Nevertheless, there is an increasing interest in generalizing convolutions to the graph domain. Advances in this direction are often categorized as spectral approaches and non-spectral approaches.

On one hand, spectral approaches work with a spectral representation of the graphs and have been successfully applied in the context of node classification. In Bruna et al. (2014), the convolution operation is defined in the Fourier domain by computing the eigendecomposition of the graph Laplacian, resulting in potentially intense computations and non-spatially localized filters. These issues were addressed by subsequent works. Henaff et al. (2015) introduced a parameterization of the spectral filters with smooth coefficients in order to make them spatially localized. Later, Defferrard et al. (2016) proposed to approximate the filters by means of a Chebyshev expansion of the graph Laplacian, removing the need to compute the eigenvectors of the Laplacian and yielding spatially localized filters. Finally, Kipf & Welling (2017) simplified the previous method by restricting the filters to operate in a 1-step neighborhood around each node. However, in all of the aforementioned spectral approaches, the learned filters depend on the Laplacian eigenbasis, which depends on the graph structure. Thus, a model trained on a specific structure can not be directly applied to a graph with a different structure.

On the other hand, we have non-spectral approaches (Duvenaud et al., 2015; Atwood & Towsley, 2016; Hamilton et al., 2017), which define convolutions directly on the graph, operating on groups of spatially close neighbors. One of the challenges of these approaches is to define an operator which works with different sized neighborhoods and maintains the weight sharing property of CNNs. In some cases, this requires learning a specific weight matrix for each node degree (Duvenaud et al., 2015), using the powers of a transition matrix to define the neighborhood while learning weights for each input channel and neighborhood degree (Atwood & Towsley, 2016), or extracting and normalizing neighborhoods containing a fixed number of nodes (Niepert et al., 2016). Monti et al. (2016) presented mixture model CNNs (MoNet), a spatial approach which provides a unified generalization of CNN architectures to graphs. More recently, Hamilton et al. (2017) introduced GraphSAGE, a method for computing node representations in an *inductive* manner. This technique operates by sampling a fixed-size neighborhood of each node, and then performing a specific aggregator over it (such as the mean over all the sampled neighbors' feature vectors, or the result of feeding them through a recurrent neural network). This approach has yielded impressive performance across several large-scale inductive benchmarks.

Attention mechanisms have become almost a *de facto* standard in many sequence-based tasks (Bahdanau et al., 2015; Gehring et al., 2016). One of the benefits of attention mechanisms is that they allow for dealing with variable sized inputs, focusing on the most relevant parts of the input to make decisions. When an attention mechanism is used to compute a representation of a single sequence, it is commonly referred to as *self-attention* or *intra-attention*. Together with Recurrent Neural Networks (RNNs) or convolutions, self-attention has proven to be useful for tasks such as machine reading (Cheng et al., 2016) and learning sentence representations (Lin et al., 2017). However, Vaswani et al. (2017) showed that not only *self-attention* can improve a method based on RNNs or convolutions, but also that it is sufficient for constructing a powerful model obtaining *state-of-the-art* performance on the machine translation task.

Inspired by this recent work, we introduce an attention-based architecture to perform node classification of graph-structured data. The idea is to compute the hidden representations of each node in the graph, by attending over its neighbors, following a *self-attention* strategy. The attention architecture has several interesting properties: (1) the operation is efficient, since it is parallelizable across node-neighbor pairs; (2) it can be applied to graph nodes having different degrees by specifying arbitrary weights to the neighbors; and (3) the model is directly applicable to *inductive* learning problems, including tasks where the model has to generalize to completely unseen graphs. We validate the proposed approach on four challenging benchmarks: *Cora*, *Citeseer* and *Pubmed* citation networks as well as an inductive *protein-protein interaction* dataset, achieving or matching state-of-the-art results that highlight the potential of attention-based models when dealing with arbitrarily structured graphs.

It is worth noting that, as Kipf & Welling (2017) and Atwood & Towsley (2016), our work can also be reformulated as a particular instance of MoNet (Monti et al., 2016). Moreover, our approach of sharing a neural network computation across edges is reminiscent of the formulation of relational networks (Santoro et al., 2017) and VAIN (Hoshen, 2017), wherein relations between objects or agents are aggregated pair-wise, by employing a shared mechanism. Similarly, our proposed attention model can be connected to the works by Duan et al. (2017) and Denil et al. (2017), which use a neighborhood attention operation to compute attention coefficients between different objects in an environment. Other related approaches include locally linear embedding (LLE) (Roweis & Saul, 2000) and memory networks (Weston et al., 2014). LLE selects a fixed number of neighbors around each data point, and learns a weight coefficient for each neighbor to reconstruct each point as a weighted sum of its neighbors. A second optimization step extracts the point's feature embedding. Memory networks also share some connections with our work, in particular, if we interpret the neighborhood of a node as the memory, which is used to compute the node features by attending over its values, and then is updated by storing the new features in the same position.

## 2 GAT ARCHITECTURE

In this section, we will present the building block layer used to construct arbitrary graph attention networks (through stacking this layer), and directly outline its theoretical and practical benefits and limitations compared to prior work in the domain of neural graph processing.

### 2.1 GRAPH ATTENTIONAL LAYER

We will start by describing a single *graph attentional layer*, as the sole layer utilized throughout all of the GAT architectures used in our experiments. The particular attentional setup utilized by us closely follows the work of Bahdanau et al. (2015)—but the framework is agnostic to the particular choice of attention mechanism.

The input to our layer is a set of node features, $\mathbf{h} = \{\vec{h}_1, \vec{h}_2, \ldots, \vec{h}_N\}, \vec{h}_i \in \mathbb{R}^F$, where $N$ is the number of nodes, and $F$ is the number of features in each node. The layer produces a new set of node features (of potentially different cardinality $F'$), $\mathbf{h}' = \{\vec{h}'_1, \vec{h}'_2, \ldots, \vec{h}'_N\}, \vec{h}'_i \in \mathbb{R}^{F'}$, as its output.

In order to obtain sufficient expressive power to transform the input features into higher-level features, at least one learnable linear transformation is required. To that end, as an initial step, a shared linear transformation, parametrized by a *weight matrix*, $\mathbf{W} \in \mathbb{R}^{F' \times F}$, is applied to every node. We then perform *self-attention* on the nodes—a shared attentional mechanism $a : \mathbb{R}^{F'} \times \mathbb{R}^{F'} \to \mathbb{R}$ computes *attention coefficients*

$$e_{ij} = a(\mathbf{W}\vec{h}_i, \mathbf{W}\vec{h}_j) \tag{1}$$

that indicate the *importance* of node $j$'s features to node $i$. In its most general formulation, the model allows every node to attend on every other node, *dropping all structural information*. We inject the graph structure into the mechanism by performing *masked attention*—we only compute $e_{ij}$ for nodes $j \in \mathcal{N}_i$, where $\mathcal{N}_i$ is some *neighborhood* of node $i$ in the graph. In all our experiments, these will be exactly the first-order neighbors of $i$ (including $i$). To make coefficients easily comparable across different nodes, we normalize them across all choices of $j$ using the softmax function:

$$\alpha_{ij} = \text{softmax}_j(e_{ij}) = \frac{\exp(e_{ij})}{\sum_{k \in \mathcal{N}_i} \exp(e_{ik})}. \tag{2}$$

In our experiments, the attention mechanism $a$ is a single-layer feedforward neural network, parametrized by a weight vector $\vec{\mathbf{a}} \in \mathbb{R}^{2F'}$, and applying the LeakyReLU nonlinearity (with negative input slope $\alpha = 0.2$). Fully expanded out, the coefficients computed by the attention mechanism (illustrated by Figure 1 (left)) may then be expressed as:

$$\alpha_{ij} = \frac{\exp\left(\text{LeakyReLU}\left(\vec{\mathbf{a}}^T[\mathbf{W}\vec{h}_i \| \mathbf{W}\vec{h}_j]\right)\right)}{\sum_{k \in \mathcal{N}_i} \exp\left(\text{LeakyReLU}\left(\vec{\mathbf{a}}^T[\mathbf{W}\vec{h}_i \| \mathbf{W}\vec{h}_k]\right)\right)} \tag{3}$$

where $\cdot^T$ represents transposition and $\|$ is the concatenation operation.

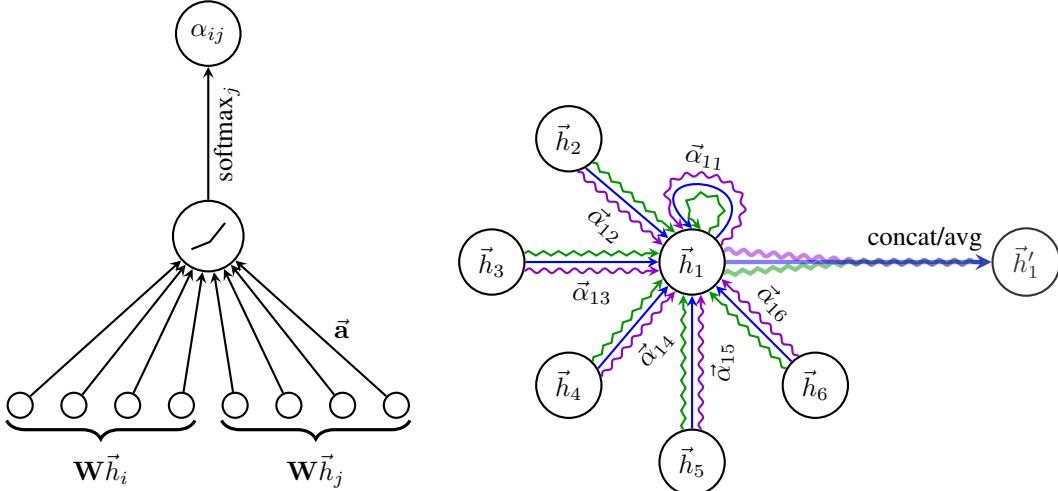

Figure 1: **Left:** The attention mechanism $a(\mathbf{W}\vec{h}_i, \mathbf{W}\vec{h}_j)$ employed by our model, parametrized by a weight vector $\vec{\mathbf{a}} \in \mathbb{R}^{2F'}$, applying a LeakyReLU activation. **Right:** An illustration of multi-head attention (with $K = 3$ heads) by node 1 on its neighborhood. Different arrow styles and colors denote independent attention computations. The aggregated features from each head are concatenated or averaged to obtain $\vec{h}'_1$.

Once obtained, the normalized attention coefficients are used to compute a linear combination of the features corresponding to them, to serve as the final output features for every node (after potentially applying a nonlinearity, $\sigma$):

$$\vec{h}'_i = \sigma\left(\sum_{j \in \mathcal{N}_i} \alpha_{ij} \mathbf{W}\vec{h}_j\right). \tag{4}$$

To stabilize the learning process of self-attention, we have found extending our mechanism to employ *multi-head attention* to be beneficial, similarly to Vaswani et al. (2017). Specifically, $K$ independent attention mechanisms execute the transformation of Equation 4, and then their features are concatenated, resulting in the following output feature representation:

$$\vec{h}'_i = \overset{K}{\underset{k=1}{\Big\|}} \sigma\left(\sum_{j \in \mathcal{N}_i} \alpha^k_{ij} \mathbf{W}^k \vec{h}_j\right) \tag{5}$$

where $\|$ represents concatenation, $\alpha^k_{ij}$ are normalized attention coefficients computed by the $k$-th attention mechanism ($a^k$), and $\mathbf{W}^k$ is the corresponding input linear transformation's weight matrix. Note that, in this setting, the final returned output, $\mathbf{h}'$, will consist of $KF'$ features (rather than $F'$) for each node.

Specially, if we perform multi-head attention on the final (prediction) layer of the network, concatenation is no longer sensible—instead, we employ *averaging*, and delay applying the final nonlinearity (usually a softmax or logistic sigmoid for classification problems) until then:

$$\vec{h}'_i = \sigma\left(\frac{1}{K}\sum_{k=1}^{K}\sum_{j \in \mathcal{N}_i} \alpha^k_{ij} \mathbf{W}^k \vec{h}_j\right) \tag{6}$$

The aggregation process of a multi-head graph attentional layer is illustrated by Figure 1 (right).

## 2.2 COMPARISONS TO RELATED WORK

The graph attentional layer described in subsection 2.1 directly addresses several issues that were present in prior approaches to modelling graph-structured data with neural networks:

- Computationally, it is highly efficient: the operation of the self-attentional layer can be parallelized across all edges, and the computation of output features can be parallelized across all nodes. No eigendecompositions or similar computationally intensive matrix operations are required. The time complexity of a single GAT attention head computing $F'$ features may be expressed as $O(|V|FF' + |E|F')$, where $F$ is the number of input features, and $|V|$ and $|E|$ are the numbers of nodes and edges in the graph, respectively. This complexity is on par with the baseline methods such as Graph Convolutional Networks (GCNs) (Kipf & Welling, 2017). Applying multi-head attention multiplies the storage and parameter requirements by a factor of $K$, while the individual heads' computations are fully independent and can be parallelized.

- As opposed to GCNs, our model allows for (implicitly) assigning *different importances* to nodes of a same neighborhood, enabling a leap in model capacity. Furthermore, analyzing the learned attentional weights may lead to benefits in interpretability, as was the case in the machine translation domain (e.g. the qualitative analysis of Bahdanau et al. (2015)).

- The attention mechanism is applied in a shared manner to all edges in the graph, and therefore it does not depend on upfront access to the global graph structure or (features of) all of its nodes (a limitation of many prior techniques). This has several desirable implications:

  - The graph is not required to be undirected (we may simply leave out computing $\alpha_{ij}$ if edge $j \to i$ is not present).

  - It makes our technique directly applicable to *inductive* learning—including tasks where the model is evaluated on graphs that are *completely unseen* during training.

- The recently published inductive method of Hamilton et al. (2017) samples a *fixed-size neighborhood* of each node, in order to keep its computational footprint consistent; this does not allow it access to the entirety of the neighborhood while performing inference. Moreover, this technique achieved some of its strongest results when an LSTM (Hochreiter & Schmidhuber, 1997)-based neighborhood aggregator is used. This assumes the existence of a consistent sequential node ordering across neighborhoods, and the authors have rectified it by consistently feeding randomly-ordered sequences to the LSTM. Our technique does not suffer from either of these issues—it works with the entirety of the neighborhood (at the expense of a variable computational footprint, which is still on-par with methods like the GCN), and does not assume any ordering within it.

- As mentioned in Section 1, GAT can be reformulated as a particular instance of MoNet (Monti et al., 2016). More specifically, setting the pseudo-coordinate function to be $u(x, y) = f(x)\|f(y)$, where $f(x)$ represent (potentially MLP-transformed) features of node $x$ and $\|$ is concatenation; and the weight function to be $w_j(u) = \text{softmax}(\text{MLP}(u))$ (with the softmax performed over the entire neighborhood of a node) would make MoNet's patch operator similar to ours. Nevertheless, one should note that, in comparison to previously considered MoNet instances, our model uses node features for similarity computations, rather than the node's structural properties (which would assume knowing the graph structure upfront).

We were able to produce a version of the GAT layer that leverages *sparse* matrix operations, reducing the storage complexity to linear in the number of nodes and edges and enabling the execution of GAT models on larger graph datasets. However, the tensor manipulation framework we used only supports sparse matrix multiplication for rank-2 tensors, which limits the batching capabilities of the layer as it is currently implemented (especially for datasets with multiple graphs). Appropriately addressing this constraint is an important direction for future work. Depending on the regularity of the graph structure in place, GPUs may not be able to offer major performance benefits compared to CPUs in these sparse scenarios. It should also be noted that the size of the "receptive field" of our model is upper-bounded by the depth of the network (similarly as for GCN and similar models). Techniques such as skip connections (He et al., 2016) could be readily applied for appropriately extending the depth, however. Lastly, parallelization across all the graph edges, especially in a distributed manner, may involve a lot of redundant computation, as the neighborhoods will often highly overlap in graphs of interest.

Table 1: Summary of the datasets used in our experiments.

|  | **Cora** | **Citeseer** | **Pubmed** | **PPI** |
|---|---|---|---|---|
| **Task** | Transductive | Transductive | Transductive | Inductive |
| **# Nodes** | 2708 (1 graph) | 3327 (1 graph) | 19717 (1 graph) | 56944 (24 graphs) |
| **# Edges** | 5429 | 4732 | 44338 | 818716 |
| **# Features/Node** | 1433 | 3703 | 500 | 50 |
| **# Classes** | 7 | 6 | 3 | 121 (multilabel) |
| **# Training Nodes** | 140 | 120 | 60 | 44906 (20 graphs) |
| **# Validation Nodes** | 500 | 500 | 500 | 6514 (2 graphs) |
| **# Test Nodes** | 1000 | 1000 | 1000 | 5524 (2 graphs) |

## 3 EVALUATION

We have performed comparative evaluation of GAT models against a wide variety of strong baselines and previous approaches, on four established graph-based benchmark tasks (transductive as well as inductive), achieving or matching state-of-the-art performance across all of them. This section summarizes our experimental setup, results, and a brief qualitative analysis of a GAT model's extracted feature representations.

### 3.1 DATASETS

**Transductive learning**   We utilize three standard citation network benchmark datasets—Cora, Citeseer and Pubmed (Sen et al., 2008)—and closely follow the transductive experimental setup of Yang et al. (2016). In all of these datasets, nodes correspond to documents and edges to (undirected) citations. Node features correspond to elements of a bag-of-words representation of a document. Each node has a class label. We allow for only 20 nodes per class to be used for training—however, honoring the transductive setup, the training algorithm has access to all of the nodes' feature vectors. The predictive power of the trained models is evaluated on 1000 test nodes, and we use 500 additional nodes for validation purposes (the same ones as used by Kipf & Welling (2017)). The Cora dataset contains 2708 nodes, 5429 edges, 7 classes and 1433 features per node. The Citeseer dataset contains 3327 nodes, 4732 edges, 6 classes and 3703 features per node. The Pubmed dataset contains 19717 nodes, 44338 edges, 3 classes and 500 features per node.

**Inductive learning**   We make use of a protein-protein interaction (PPI) dataset that consists of graphs corresponding to different human tissues (Zitnik & Leskovec, 2017). The dataset contains 20 graphs for training, 2 for validation and 2 for testing. Critically, testing graphs remain *completely unobserved* during training. To construct the graphs, we used the preprocessed data provided by Hamilton et al. (2017). The average number of nodes per graph is 2372. Each node has 50 features that are composed of positional gene sets, motif gene sets and immunological signatures. There are 121 labels for each node set from gene ontology, collected from the Molecular Signatures Database (Subramanian et al., 2005), and a node can possess several labels simultaneously.

An overview of the interesting characteristics of the datasets is given in Table 1.

### 3.2 STATE-OF-THE-ART METHODS

**Transductive learning**   For transductive learning tasks, we compare against the same strong baselines and state-of-the-art approaches as specified in Kipf & Welling (2017). This includes label propagation (LP) (Zhu et al., 2003), semi-supervised embedding (SemiEmb) (Weston et al., 2012), manifold regularization (ManiReg) (Belkin et al., 2006), skip-gram based graph embeddings (Deep-Walk) (Perozzi et al., 2014), the iterative classification algorithm (ICA) (Lu & Getoor, 2003) and Planetoid (Yang et al., 2016). We also directly compare our model against GCNs (Kipf & Welling, 2017), as well as graph convolutional models utilising higher-order Chebyshev filters (Defferrard et al., 2016), and the MoNet model presented in Monti et al. (2016).

**Inductive learning**  For the inductive learning task, we compare against the four different supervised GraphSAGE inductive methods presented in Hamilton et al. (2017). These provide a variety of approaches to aggregating features within a sampled neighborhood: GraphSAGE-GCN (which extends a graph convolution-style operation to the inductive setting), GraphSAGE-mean (taking the elementwise mean value of feature vectors), GraphSAGE-LSTM (aggregating by feeding the neighborhood features into an LSTM) and GraphSAGE-pool (taking the elementwise maximization operation of feature vectors transformed by a shared nonlinear multilayer perceptron). The other transductive approaches are either completely inappropriate in an inductive setting or assume that nodes are incrementally added to a single graph, making them unusable for the setup where test graphs are completely unseen during training (such as the PPI dataset).

Additionally, for both tasks we provide the performance of a per-node shared multilayer perceptron (MLP) classifier (that does not incorporate graph structure at all).

### 3.3 EXPERIMENTAL SETUP

**Transductive learning**  For the transductive learning tasks, we apply a two-layer GAT model. Its architectural hyperparameters have been optimized on the Cora dataset and are then reused for Citeseer. The first layer consists of $K = 8$ attention heads computing $F' = 8$ features each (for a total of 64 features), followed by an exponential linear unit (ELU) (Clevert et al., 2016) nonlinearity. The second layer is used for classification: a single attention head that computes $C$ features (where $C$ is the number of classes), followed by a softmax activation. For coping with the small training set sizes, regularization is liberally applied within the model. During training, we apply $L_2$ regularization with $\lambda = 0.0005$. Furthermore, dropout (Srivastava et al., 2014) with $p = 0.6$ is applied to both layers' inputs, as well as *to the normalized attention coefficients* (critically, this means that at each training iteration, each node is exposed to a stochastically sampled neighborhood). Similarly as observed by Monti et al. (2016), we found that Pubmed's training set size (60 examples) required slight changes to the GAT architecture: we have applied $K = 8$ output attention heads (instead of one), and strengthened the $L_2$ regularization to $\lambda = 0.001$. Otherwise, the architecture matches the one used for Cora and Citeseer.

**Inductive learning**  For the inductive learning task, we apply a three-layer GAT model. Both of the first two layers consist of $K = 4$ attention heads computing $F' = 256$ features (for a total of 1024 features), followed by an ELU nonlinearity. The final layer is used for (multi-label) classification: $K = 6$ attention heads computing 121 features each, that are averaged and followed by a logistic sigmoid activation. The training sets for this task are sufficiently large and we found no need to apply $L_2$ regularization or dropout—we have, however, successfully employed skip connections (He et al., 2016) across the intermediate attentional layer. We utilize a batch size of 2 graphs during training. To strictly evaluate the benefits of applying an attention mechanism in this setting (i.e. comparing with a near GCN-equivalent model), we also provide the results when a *constant attention mechanism*, $a(x, y) = 1$, is used, with the same architecture—this will assign the same weight to every neighbor.

Both models are initialized using Glorot initialization (Glorot & Bengio, 2010) and trained to minimize cross-entropy on the training nodes using the Adam SGD optimizer (Kingma & Ba, 2014) with an initial learning rate of 0.01 for Pubmed, and 0.005 for all other datasets. In both cases we use an early stopping strategy on both the cross-entropy loss and accuracy (transductive) or micro-$F_1$ (inductive) score on the validation nodes, with a patience of 100 epochs[1].

### 3.4 RESULTS

The results of our comparative evaluation experiments are summarized in Tables 2 and 3.

For the transductive tasks, we report the mean classification accuracy (with standard deviation) on the test nodes of our method after 100 runs, and reuse the metrics already reported in Kipf & Welling (2017) and Monti et al. (2016) for state-of-the-art techniques. Specifically, for the Chebyshev filter-based approach (Defferrard et al., 2016), we provide the maximum reported performance for filters of orders $K = 2$ and $K = 3$. In order to fairly assess the benefits of the attention mechanism, we further evaluate a GCN model that computes 64 hidden features, attempting both the ReLU and

---

[1]Our implementation of the GAT layer may be found at: `https://github.com/PetarV-/GAT`.

Table 2: Summary of results in terms of classification accuracies, for Cora, Citeseer and Pubmed. GCN-64* corresponds to the best GCN result computing 64 hidden features (using ReLU or ELU).

| | *Transductive* | | |
|---|---|---|---|
| **Method** | **Cora** | **Citeseer** | **Pubmed** |
| MLP | 55.1% | 46.5% | 71.4% |
| ManiReg (Belkin et al., 2006) | 59.5% | 60.1% | 70.7% |
| SemiEmb (Weston et al., 2012) | 59.0% | 59.6% | 71.7% |
| LP (Zhu et al., 2003) | 68.0% | 45.3% | 63.0% |
| DeepWalk (Perozzi et al., 2014) | 67.2% | 43.2% | 65.3% |
| ICA (Lu & Getoor, 2003) | 75.1% | 69.1% | 73.9% |
| Planetoid (Yang et al., 2016) | 75.7% | 64.7% | 77.2% |
| Chebyshev (Defferrard et al., 2016) | 81.2% | 69.8% | 74.4% |
| GCN (Kipf & Welling, 2017) | 81.5% | 70.3% | **79.0%** |
| MoNet (Monti et al., 2016) | $81.7 \pm 0.5\%$ | — | $78.8 \pm 0.3\%$ |
| GCN-64* | $81.4 \pm 0.5\%$ | $70.9 \pm 0.5\%$ | **79.0** $\pm 0.3\%$ |
| **GAT** (ours) | **83.0** $\pm 0.7\%$ | **72.5** $\pm 0.7\%$ | **79.0** $\pm 0.3\%$ |

Table 3: Summary of results in terms of micro-averaged $F_1$ scores, for the PPI dataset. GraphSAGE* corresponds to the best GraphSAGE result we were able to obtain by just modifying its architecture. Const-GAT corresponds to a model with the same architecture as GAT, but with a constant attention mechanism (assigning same importance to each neighbor; GCN-like inductive operator).

| | *Inductive* |
|---|---|
| **Method** | **PPI** |
| Random | 0.396 |
| MLP | 0.422 |
| GraphSAGE-GCN (Hamilton et al., 2017) | 0.500 |
| GraphSAGE-mean (Hamilton et al., 2017) | 0.598 |
| GraphSAGE-LSTM (Hamilton et al., 2017) | 0.612 |
| GraphSAGE-pool (Hamilton et al., 2017) | 0.600 |
| GraphSAGE* | 0.768 |
| Const-GAT (ours) | $0.934 \pm 0.006$ |
| **GAT** (ours) | **0.973** $\pm 0.002$ |

ELU activation, and reporting (as GCN-64*) the better result after 100 runs (which was the ReLU in all three cases).

For the inductive task, we report the micro-averaged $F_1$ score on the nodes of the two unseen test graphs, averaged after 10 runs, and reuse the metrics already reported in Hamilton et al. (2017) for the other techniques. Specifically, as our setup is supervised, we compare against the supervised GraphSAGE approaches. To evaluate the benefits of aggregating across the entire neighborhood, we further provide (as GraphSAGE*) the best result we were able to achieve with GraphSAGE by just modifying its architecture (this was with a three-layer GraphSAGE-LSTM with [512, 512, 726] features computed in each layer and 128 features used for aggregating neighborhoods). Finally, we report the 10-run result of our constant attention GAT model (as Const-GAT), to fairly evaluate the benefits of the attention mechanism against a GCN-like aggregation scheme (with the same architecture).

Our results successfully demonstrate state-of-the-art performance being achieved or matched across all four datasets—in concordance with our expectations, as per the discussion in Section 2.2. More specifically, we are able to improve upon GCNs by a margin of 1.5% and 1.6% on Cora and Citeseer, respectively, suggesting that assigning different weights to nodes of a same neighborhood may be beneficial. It is worth noting the improvements achieved on the PPI dataset: Our GAT model

improves by 20.5% w.r.t. the best GraphSAGE result we were able to obtain, demonstrating that our model has the potential to be applied in inductive settings, and that larger predictive power can be leveraged by observing the entire neighborhood. Furthermore, it improves by 3.9% w.r.t. Const-GAT (the identical architecture with constant attention mechanism), once again directly demonstrating the significance of being able to assign different weights to different neighbors.

The effectiveness of the learned feature representations may also be investigated qualitatively—and for this purpose we provide a visualization of the t-SNE (Maaten & Hinton, 2008)-transformed feature representations extracted by the first layer of a GAT model pre-trained on the Cora dataset (Figure 2). The representation exhibits discernible clustering in the projected 2D space. Note that these clusters correspond to the seven labels of the dataset, verifying the model's discriminative power across the seven topic classes of Cora. Additionally, we visualize the relative strengths of the normalized attention coefficients (averaged across all eight attention heads). Properly interpreting these coefficients (as performed by e.g. Bahdanau et al. (2015)) will require further domain knowledge about the dataset under study, and is left for future work.

## 4 CONCLUSIONS

We have presented graph attention networks (GATs), novel convolution-style neural networks that operate on graph-structured data, leveraging masked self-attentional layers. The graph attentional layer utilized throughout these networks is computationally efficient (does not require computationally intensive matrix operations, and is parallelizable across all nodes in the graph), allows for (implicitly) assigning different importances to different nodes within a neighborhood while dealing with different sized neighborhoods, and does not depend on knowing the entire graph structure upfront—thus addressing many of the theoretical issues with previous spectral-based approaches. Our models leveraging attention have successfully achieved or matched state-of-the-art performance across four well-established node classification benchmarks, both transductive and inductive (especially, with completely unseen graphs used for testing).

There are several potential improvements and extensions to graph attention networks that could be addressed as future work, such as overcoming the practical problems described in subsection 2.2 to be able to handle larger batch sizes. A particularly interesting research direction would be taking advantage of the attention mechanism to perform a thorough analysis on the model interpretability.

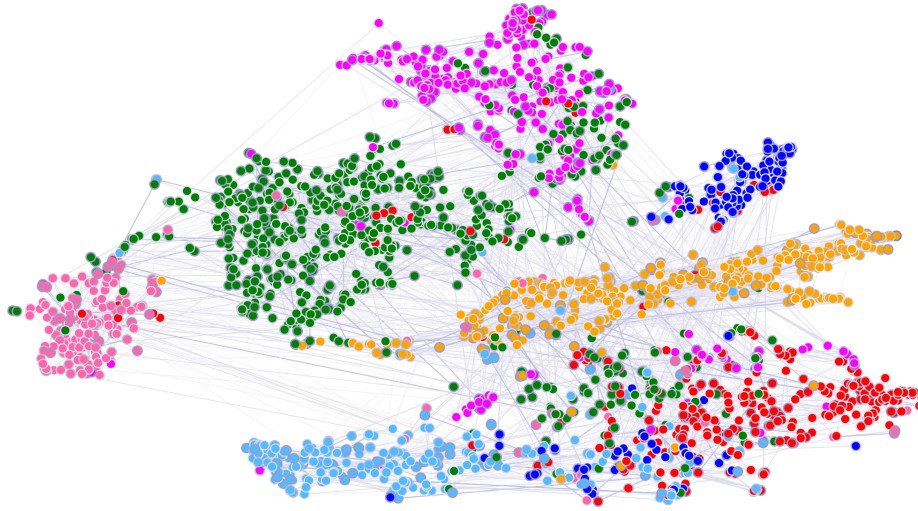

Figure 2: A t-SNE plot of the computed feature representations of a pre-trained GAT model's first hidden layer on the Cora dataset. Node colors denote classes. Edge thickness indicates aggregated normalized attention coefficients between nodes $i$ and $j$, across all eight attention heads $(\sum_{k=1}^{K} \alpha_{ij}^k + \alpha_{ji}^k)$.

Moreover, extending the method to perform graph classification instead of node classification would also be relevant from the application perspective. Finally, extending the model to incorporate edge features (possibly indicating relationship among nodes) would allow us to tackle a larger variety of problems.

ACKNOWLEDGEMENTS

The authors would like to thank the developers of TensorFlow (Abadi et al., 2015). PV and PL have received funding from the European Union's Horizon 2020 research and innovation programme PROPAG-AGEING under grant agreement No 634821. We further acknowledge the support of the following agencies for research funding and computing support: CIFAR, Canada Research Chairs, Compute Canada and Calcul Québec, as well as NVIDIA for the generous GPU support. Special thanks to: Benjamin Day and Fabian Jansen for kindly pointing out issues in a previous iteration of the paper; Michał Drożdżal for useful discussions, feedback and support; and Gaétan Marceau for reviewing the paper prior to submission.

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
