# OpenReview forum: "Graph Attention Networks"
_ICLR.cc/2018/Conference — Accept (Poster)_

### Official Review · AnonReviewer2 · 2017-11-27
**Very interesting work, but the graph structure is not fully exploited**

**Rating:** 7
**Confidence:** 5

**Review:**

The paper introduces a neural network architecture to operate on graph-structured
data named Graph Attention Networks.
Key components are an attention layer and the possibility to learn how to
weight different nodes in the neighborhood without requiring spectral decompositions
which are costly to be computed.

I found the paper clearly written and very well presented. I want to thank
the author for actively participating in the discussions and in clarifying already
many of the details that I was missing.

As also reported in the comments by T. Kipf I found the lack of comparison to previous
works on attention and on constructions of NN for graph data are missing.
In particular MoNet seems a more general framework, using features to compute node
similarity is another way to specify the "coordinate system" for convolution.
I would argue that in many cases the graph is given and that one would have
to exploit its structure rather than the simple first order neighbors structure.

I feel, in fact, that the paper deals mainly with "localized metric-learning" rather than
using the information in the graph itself. There is no
explicit usage of the graph beyond the selection of the local neighborhood.
In many ways when I first read it I though it would be a modified version of
memory networks (which have not been cited). Sec. 2.1 is basically describing
a way to learn a matrix W so that the attention layer produces the weights to be
used for convolution, or the relative coordinate system, which is to me a
memory network like construction, where the memory is given by the neighborhood.

I find the idea to use the multi-head attention very interesting, but one should
consider the increase in number of parameters in the experimental section.

I agree that the proposed method is computationally efficient but the authors
should keep in mind that parallelizing across all edges involves lot of redundant
copies (e.g. in a distributed system) as the neighborhoods highly overlap, at
least for interesting graphs.

The advantage with respect to methods that try to use LSTM in this domain
in a naive manner is clear, however the similarity function (attention) in this
work could be interpreted as the variable dictating the visit ordering.

The authors seem to emphasize the use of GPU as the best way to scale their work
but I tend to think that when nodes have varying degrees they would be highly
unused. Main reason why they are widely used now is due to the structure in the
representation of convolutional operations.
Also in case of sparse data GPUs are not the best alternative.

Experiments are very well described and performed, however as explained earlier
some comparisons are needed.
An interesting experiment could be to use the attention weights as adjacency
matrix for GCN.

Overall I liked the paper and the presentation, I think it is a simple yet
effective way of dealing with graph structure data. However, I think that in
many interesting cases the graph structure is relevant and cannot be used
just to get the neighboring nodes (e.g. in social network analysis).

---

> ### Author Response · Authors · 2017-12-19
> **Reply to AnonReviewer2**
>
> First of all, thank you very much for your thorough review, and for the variety of useful pointers within it! Please refer to our global comment above for a list of all revisions we have applied to the paper---we are hopeful that they have addressed your comments appropriately.
>
> We have now added all the references to attention-like constructions (such as MoNet and neighbourhood attention) to our related work, as well as memory networks (see Section 1, paragraphs 6 and 9; also Section 2.2, bullet point 5). We fully agree with your comments about the increase in parameter count with multi-head attention, computational redundancy, and comparative advantages of GPUs in this domain, and have explicitly added them as remarks to our model’s analysis (in Section 2.2, bullet point 1 and paragraph 2).
>
> While we agree that the graph structure is given in many interesting cases, in our approach we specifically sought to produce an operator explicitly capable of solving inductive problems (which appear often, e.g., in the biomedical domain, where the method needs to be able to generalise to new structures). A potential way of reconciling this when a graph structure is provided is to combine GAT-like and spectral layers in the same architecture.
>
> Further experiments (as discussed by us in all previous comments) have also been performed and are now explicitly listed in the paper’s Results section (please see Tables 2 and 3 for a summary). We have also attempted to use the GAT coefficients as the aggregation matrix for GCNs (both in an averaged and multi-head manner)---but found that there were no clear performance changes compared to using the Laplacian.
>
> We thank you once again for your review, which has definitely helped make our paper’s contributions stronger!

---

### Official Review · AnonReviewer3 · 2017-11-29
**Good basic idea with several weaknesses in the technical exposition and the experiments**

**Rating:** 5
**Confidence:** 4

**Review:**

This is a paper about learning vector representations for the nodes of a graph. These embeddings can be used in downstream tasks the most common of which is node classification.

Several existing approaches have been proposed in recent years. The authors provide a fair and almost comprehensive  discussion of state of the art approaches. There are a couple of exception that have already been mentioned in a comment from Thomas Kipf and Michael Bronstein. A more precise discussion of the differences between existing approaches (especially MoNets) should be a crucial addition to the paper. You provide such a  comparison in your answer to Michael's comment. To me, the comparison makes sense but it also shows that the ideas presented here are less novel than they might initially seem. The proposed method introduces two forms of (simple) attention. Nothing groundbreaking here but still interesting enough and well explained. It might also be a good idea to compare your method to something like LLE (locally linear embedding). LLE also learns a weight for each of neighbors of a node and computes the embedding as a weighted average of the neighbor embeddings according to these weights. Your approach is different since it is learned end-to-end (not in two separate steps) and because it is applicable to arbitrary graphs (not just graphs where every node has exactly k neighbors as in LLE). Still, something to relate to.

Please take a look at the comment by Fabian Jansen. I think he is on to something. It seems that the attention weight (from i to j) in the end is only a normalization operation that doesn't take the embedding of node i into account.

There are two  issues with the experiments.

First, you don't report results on Pubmed because your method didn't scale. Considering that Pubmed has less than 20,000 nodes this shows a clear weakness of your approach. You write (in an answer to a comment) that it *should* be parallelizable but somehow you didn't make it work. We have to, however, evaluate the approach on what it is able to do at the moment. Having a complexity that is quadratic in the number of nodes is terrible and one of the major reasons learning with graphs has moved from kernels to neural approaches. While it is great that you acknowledge this openly as a weakness, it is currently not possible to claim that your method scales to even moderately sized graphs.

Second, the experimental set-up on the Cora and Citeseer data sets should be properly randomized. As Thomas pointed out, for graph data the variance can be quite high. For some split the method might perform really well and less well for others. In your answer titled "Requested clarifications" to a different comment you provide numbers randomized over 10 runs. Did you randomize the parameter initialization only or also the the train/val/test splits? If you did the latter, this seems reasonable. In Kipf et al.'s GCN paper this is what was done (not over 100 splits as some other commenter claimed. The average over 100 runs  pertained to the ICA method only.)

---

> ### Author Response · Authors · 2017-12-19
> **Reply to AnonReviewer3**
>
> Thank you very much for your detailed review! Please refer to our global comment above for a list of all revisions we have applied to the paper---we are hopeful that they have addressed your comments appropriately.
>
> Fabian has indeed correctly identified that half of our attention weights were spurious. We have now rectified this by applying a simple nonlinearity (the LeakyReLU) prior to normalising, and anticipated that its application would provide better performance to the model on the PPI dataset (which has a large number of training nodes). Indeed, we noticed no discernible change on Cora and Citeseer, but an increase in F1-score on PPI (now at 0.973 +- 0.002 after 10 runs; previously, as given in our reply to one of the comments below, it was 0.952 +- 0.006). The new results may be found in Tables 2 and 3.
>
> In the meantime, we have been successful at leveraging TensorFlow’s sparse_softmax operation, and produced a sparsified version of the GAT layer. We are happy to provide results on Pubmed, and they are now given in the revised version of the paper (see Table 2 for a summary). We were able to match state-of-the-art level performance of MoNet and GCN (at 79.0 +- 0.3% after 100 runs). Similarly to the MoNet paper authors, we had to revise the GAT architecture slightly to accommodate Pubmed’s extremely small training set size (of 60 examples), and this is clearly remarked in our experimental setup (Section 3.3).
>
> Finally, quoting directly from the work of Kipf and Welling:
>
> “We trained and tested our model on the same dataset splits as in Yang et al. (2016) and report mean accuracy of 100 runs with random weight initializations.”
>
> This implies that the splits were not randomised in the result reported by the GCN paper (specifically, the one used to compare with other baseline approaches), but only the model initialisation---and this is exactly what we do as well. We, in fact, use exactly the code provided by Thomas Kipf at https://github.com/tkipf/gcn/blob/master/gcn/utils.py#L24 to load the dataset splits.
>
> We have added all the required references to MoNet and LLE (and many other pieces of related work) in the revised version (Section 1, paragraphs 6 and 9; also Section 2.2, bullet point 5) - thank you for pointing out LLE to us, which is an interesting and relevant piece of related work!
>
> We thank you once again for your review, which has definitely helped make our paper’s contributions stronger!

---

### Official Review · AnonReviewer4 · 2017-12-03
**Well written paper, lack of novelty**

**Rating:** 6
**Confidence:** 4

**Review:**

This paper has proposed a new method for classifying nodes of a graph. Their method can be used in both semi-supervised scenarios where the label of some of the nodes of the same graph as the graph in training is missing (Transductive) and in the scenario that the test is on a completely new graph (Inductive).
Each layer of the network consists of feature representations for all of the nodes in the Graph. A linear transformation is applied to all the features in one layer and the output of the layer is the weighted sum of the transformed neighbours (including the node). The attention logit between node i and its neighbour k is calculated by a one layer fully connected network on top of the concatenation of the transformed representation of node i and transformed representation of the neighbour k. They also can incorporate the multi-head attention mechanism and average/concatenate the output of each head.

Originality:
Authors improve upon GraphSAGE by replacing the aggregate and sampling function at each layer with an attention mechanism. However, the significance of the attention mechanism has not been studied in the experiments. For example by reporting the results when attention is turned off (1/|N_i| for every node) and only a 0-1 mask for neighbours is used. They have compared with GraphSAGE only on PPI dataset. I would change my rating if they show that the 33% gain is mainly due to the attention in compare to other hyper-parameters. [The experiments are now more informative. Thanks]
Also, in page 4 authors claim that GraphSAGE is limited because it samples a neighbourhood of each node and doesn't aggregate over all the neighbours in order to keep its computational footprint consistent. However, the current implementation of the proposed method is computationally equal to using all the vertices in GraphSAGE.

Pros:
- Interesting combination of attention and local graph representation learning.
- Well written paper. It conveys the idea clearly.
- State-of-the-art results on three datasets.

Cons:
- When comparing with spectral methods it would be better to mention that the depth of embedding propagation in this method is upper-bounded by the depth of the network. Therefore, limiting its adaptability to broader class of graph datasets.
- Explaining how attention relates to previous body of work in embedding propagation and when it would be more powerful.

---

> ### Author Response · Authors · 2017-12-19
> **Reply to AnonReviewer4**
>
> We would like to thank you for the comprehensive review! Please refer to our global comment above for a list of all revisions we have applied to the paper---we are hopeful that they have addressed your comments appropriately.
>
> Primarily, thank you for suggesting the constant-attention experiment (with 1/|Ni| coefficients)! This not only directly evaluates the significance of the attention mechanism on the inductive task, but allows for a comparison with a GCN-like inductive structure. We have successfully shown a benefit of using attention:
>
> The Const-GAT model achieved 0.934 +- 0.006 micro-F1;
> The GAT model achieved 0.973 +- 0.002 micro-F1.
>
> Which demonstrates a clear positive effect of using an attention mechanism (given that all other architectural and training properties are kept fixed across the two models). These results are clearly communicated in our revised paper now (Section 3.3 introduces the experiment in the “Inductive learning” paragraph, while the results are outlined in Table 3 and discussed in Section 3.4, paragraph 4).
>
> Our intention was not to imply that our method is computationally more efficient than GraphSAGE---only that GraphSAGE’s design decisions (sampling subsets of neighbourhoods) have potentially limiting effects on its predictive power. We have rewrote bullet point 4 in Section 2.2, to hopefully communicate this better.
>
> Lastly, we make explicit that the depth of our propagation is upper-bounded by network depth in Section 2.2, paragraph 2. We remark that GCN-like models suffer from the same issue, and that skip connections (or similar constructs) may be readily used to effectively increase the depth to desirable levels. The primary benefit of leveraging attention, as opposed to prior approaches to graph-structured feature aggregation, is being able to (implicitly) assign different importances to different neighbours, while simultaneously generalising to a wide range of degree distributions---these differences are stated in our paper in various locations (e.g. Section 1, paragraph 8; Section 2.2, bullet point 2).
>
> We thank you once again for your review, which has definitely helped make our paper’s contributions stronger!

---

### Public Comment · (anonymous) · 2017-11-02
**Clarification for the experimental setup**

In the main results in the accuracies in Table 2 and F1 scores on Table 3, are those numbers averaged over multiple training instances of the model with random initializations or are they the numbers corresponding to the best performing model? In the former case, how many random instances is it averaged over?

---

> ### Author Response · Authors · 2017-11-06
> **Requested clarifications**
>
> Thank you very much for your comment - we acknowledge that this detail about our experimental setup was not sufficiently clear in the submitted version and are more than happy to address it appropriately in a subsequent revision.
>
> We have picked the best hyperparameter configuration considering the validation score on both Cora and PPI, and then reused the Cora architectural hyperparameters on Citeseer. Once the hyperparameters were in place, the early-stopped models were then evaluated on the test set once, and the obtained results are the ones reported in the paper.
>
> We agree that reporting the averaged model performance would be useful, and we will do this in an updated version of the paper. The results after 10 runs of the same model with different random seeds are (with highlighted standard deviations):
>
> Cora: 83.0 +- 0.6 (with a maximum of 83.9%)
> Citeseer: 72.7 +- 0.7 (with a maximum of 74.2%)
> PPI: 0.952 +- 0.006 (with a maximum of 0.966)
>
> These correspond to state-of-the-art results across all three datasets.

---

### Public Comment · ~Yedid_Hoshen1 · 2017-11-06
**Attentional Multi-agent Predictive Modeling**

Interesting work!

I've done some related work, that will be presented at NIPS: https://arxiv.org/abs/1706.06122
I wonder how the two works compare?

---

> ### Author Response · Authors · 2017-11-08
> **Relationship to VAIN**
>
> Thank you for the positive feedback, as well as bringing your paper to our attention! We have found it to be very interesting related work, and will be sure to cite it in a subsequent version of our paper (most likely alongside our existing citation of the work of Santoro et al.: https://arxiv.org/abs/1706.01427 ). We highlight a few comparisons between our approaches that are worth mentioning below.
>
> We compute attention coefficients using an edge-wise mechanism, rather than a node-wise mechanism followed by an edge-wise distance metric. This is suitable for a graph setting (with neighbourhoods specified by the graph structure), because we can only evaluate this mechanism across the edges that are in the graph (easing the computational load). In a multi-agent setting (as the one explored by your paper), there may not be an immediately-obvious such structure, and this is why one has to resort to specifying interactions across all pairs of agents (at least initially, before the kind of pruning by way of k-NN could be performed). As we focus on making per-node predictions in graphs, we also found it useful for a node to attend over its own features, which your proposed model explicitly disallows. Our work also features a few stabilising additions to the attention model (to better cope with the smaller training set sizes), such as multi-head attention and dropout on the computed attention coefficients.

---

### Public Comment · ~Michael_Bronstein1 · 2017-11-06
**particular case of mixture model net?**

The model you propose looks very similar to mixture model networks (MoNet):

http://arxiv.org/pdf/1611.0840.pdf (appeared as oral at CVPR 2017)

which you did not cite.

MoNet model performed better than GCN and Chebyshev net (both of which can be considered as a particular instance thereof). What is the difference/similarity of your approach compared to MoNet?

---

> ### Author Response · Authors · 2017-11-08
> **Relationship to MoNets**
>
> Thank you very much for your comment, and pointing us to this work! MoNets are definitely a highly relevant piece of related work to ours, and therefore they will receive appropriate treatment and a citation in the subsequent revision of our paper.
>
> We find that our work can indeed be reformulated as a particular case of the MoNet framework. Namely, setting the pseudo-coordinate function to be
>
> u(x, y) = f(x) || f(y)
> (where f(x) represent (potentially MLP-transformed) features of node x, and || is concatenation)
>
> and the weight function to be
>
> w_j(u) = softmax(MLP(u))
> (with the softmax performed over the entire neighbourhood of a node)
>
> would make the patch operator similar to ours.
>
> This could be interpreted as a way of integrating the ideas of self-attentional interfaces (such as the work of Vaswani et al.: https://arxiv.org/abs/1706.03762 ) into the patch-operator framework presented by MoNet. Specially, and in comparison to the previously specified MoNet frameworks, our model uses node features for similarity computations, rather than the node's structural properties (such as their degrees in the graph). This, in combination with using a multilayer perceptron for computing the attention coefficients, allows the network more freedom in the way it chooses to express similarities between different nodes in the graph, irrespective of the local topological properties. The addition of the softmax function ensures that these coefficients will be well-behaved (and potentially probabilistically interpretable).
>
> Lastly, our work also features a few stabilising additions to the attention model (to better cope with the smaller training set sizes), such as applying dropout on the computed attention coefficients, exposing the network to a stochastically sampled neighbourhood on every iteration. Such regularisation techniques might be harder to interpret or justify when structural properties are used as pseudo-coordinates, as stochastically dropping neighbours changes e.g. the node degrees.
>
> To avoid any potential confusion for other readers of this discussion, we would like to also highlight that the arXiv link for MoNets that we referred to is: https://arxiv.org/pdf/1611.08402.pdf

---

### Public Comment · ~Thomas_N._Kipf1 · 2017-11-08
**Effect of ELU activation function and further references**

Very nicely presented work.

I was wondering how much influence the ELU activation function had on your results? It looks like all baseline models make use of ReLU for easier comparison.

In terms of datasets: did you use the same splits for Cora and Citeseer as in previous work (e.g. Kipf&Welling, ICLR2017), or did you merely use the same size of split and resample? In my experience, the choice of train/val/test splits can have a very significant impact on test performance (it is possible to get up to 84% accuracy on Cora using a lucky train/val/test split with earlier models as well).

As mentioned by others, you might want to refer to earlier work on attention mechanisms for graph neural networks or using multiple basis functions ("attention heads"), such as in the MoNet paper. Here are some references:

https://arxiv.org/abs/1611.08402  - MoNets: looks like your model is a special case of theirs, they also compare on the same kinds of tasks but avoid scalability issues by not having the softmax attention formalism
https://arxiv.org/abs/1703.07326 - Introduces "Neighborhood attention"
https://arxiv.org/abs/1706.06383 - Improved version of "Neighborhood attention"
https://arxiv.org/abs/1706.06122 - Attention mechanism in a graph neural net model for multi-agent reinforcement learning

---

> ### Author Response · Authors · 2017-11-09
> **Utilised splits, and a comment on ELU**
>
> Thank you for the kind feedback, the plethora of useful related work, and the queries!
>
> We have already noted the relationship of our work to MoNets and VAIN (as given in our replies to the authors below). The work on Neighbourhood attention is also relevant, and will also be cited appropriately alongside the related work by Santoro et al. (which we already cited in the original version). Also, the improved neighbourhood attention might hold interesting future work avenues (such as introducing an edge-wise 'message passing' network whose outputs one can attend over).
>
> We have utilised exactly the same training/validation/testing splits for Cora and Citeseer as the ones used in Kipf & Welling. This information should be already highlighted in the description of our experimental setup. In fact, for extracting the dataset we use exactly the code provided at: https://github.com/tkipf/gcn/blob/master/gcn/utils.py
>
> We have found early on in our experiments that the properties of the ELU function are convenient for simplifying the optimisation process of our method - reducing the amount of effort invested in our hyperparameter search.

---

### Public Comment · (anonymous) · 2017-11-13
**Results are inconclusive without sufficient baseline experiments**

The experiment section is not clearly indicative of what attributed to the improvement in results.

For experiments on Cora and Citeseer datasets, the authors have used the same train/test/val split as used in Kipf&Welling, ICLR'17.  Though the authors have used the same split, it is not sufficient to compare them on the original reported results from the baseline papers to indicate that the attention mechanism alone is providing improved results without analysing the
differences in architecture and experiment results. The proposed model besides the proposed attention mechanism has additional learning capacity as the model introduces an additional linear layer for input projection and has more number of hidden units per layer. Kipf's GCN has reported results with hidden units set to 16 whereas the size of the attention feature size (8*8) is 64. It is not clear how much improvement does the increased hidden size provides. It would be clear if the authors report results for the GCN and Chebyshev model with and additional input linear layer and hidden size set to 64. And also report the effect of use of elu activation functions instead of RELU as previously mentioned by Thomas Kipf in his comment.

Similarly in the inductive learning task the attention feature size is 1024 whereas the max feature size for the GraphSage models are 256. The GraphSage results are reported with partial neighborhood information from 2 hop neighbors. Whereas, in this paper the authors have used skip connections (also used in GCN) and the complete neighborhood information from 4-hop neighbors. No analysis of the effect of these two components are mentioned. It is not clear how the proposed model would perform under the same setting as in GraphSage (2-hop and partial neighborhood) or how much improvement would GraphSage obtain with skip connections and 4-hop information. On a side note, as GraphSage is not efficient to work with complete neighborhood, the authors can use Kipf's implementation of GCN and Chebyshev to report results on PPI with 4-hop complete neighborhood information and skip-connection. With these additional experiments, it would be clear how much improvement does the proposed attention mechanism exactly provides.

Further, I'm surprised that attention mechanism can provide improved results especially with Cora and Citeseer where the average degree is less than 2. These two datasets are highly homophilous. It would be useful if the authors report the mean attention score for the self edge and its neighbors.

---

> ### Author Response · Authors · 2017-11-17
> **Further baseline experimental results**
>
> First of all, thank you very much for your thorough comment and thoughts on the experimental setup!
>
> We directly quoted back the baseline results originally reported, under the assumption that appropriate hyperparameter optimisation had already been performed on them. However, we have now performed further experiments on the baseline techniques, in line with some of your recommendations, and the results of this study still point to an outperformance by GAT models. We focused on the experiments that were easily runnable without significantly modifying the codebases at https://github.com/tkipf/gcn and https://github.com/williamleif/GraphSAGE. Our findings can be summarised as follows, and will be highlighted in an updated version of the paper:
>
> Cora/Citeseer: We have trained the GCN and Chebyshev (K = 2 and K = 3) models, with a hidden size of 64, with ReLU and ELU activations, for 10 runs each. Note that we did not need to add an additional input linear layer (as suggested by the comment), given that the code at https://github.com/tkipf/gcn/blob/master/gcn/layers.py#L176 already does this.
>
> The best-performing models achieved the following mean +- std results:
>
> Cora: 81.5 +- 0.7 (Cheby2 ReLU)
> Citeseer: 71.0 +- 0.3 (Cheby3 ELU and GCN ReLU)
>
> These results are still outperformed by both our model's single-run performance (as in our paper) and 10-run performance (as in our reply to a previous comment below).
>
> PPI: Firstly, we would like to note that our model actually considers three-hop neighbourhoods (rather than four), and that the GraphSAGE models feature skip connections---in fact, our model only has one skip connection in total whereas GraphSAGE has a skip connection for every aggregation layer (the concat operation in Line 5 of Algorithm 1 in https://arxiv.org/abs/1706.02216). The authors of GraphSAGE have, in fact, highlighted that this skip connection was critical to their performance gains.
>
> In line with this, we have tested a wide range of larger GraphSAGE models with three aggregation layers, with both ReLU and ELU activations, spanning feature counts up to 1024. Specially, for the third layer we focused on feature counts of 121 and 726, as our GAT model’s final aggregation layer also acts as a classification layer, computing 6 * 121 features which are then pointwise-averaged. Some of these combinations resulted in OOM errors, with the best performing one being a GraphSAGE-LSTM model computing [512, 512, 726] features, with 128 features being used for aggregating neighbourhoods, using the ELU activation. This approach achieved a micro-F1 score of 0.648 on PPI. We have found it beneficial to let the model train for more epochs compared to the original authors' work, and were able to reach a maximal test micro-F1 score of 0.768 after doing so.
>
> This is still outperformed by a significant margin by both our single-model result (reported in the paper) and our 10-run result (reported in a reply to a previous comment below).
>
> Finally, as pointed out by the comment, we report that, for a pre-trained GAT model on Cora, the mean attentional coefficients in the hidden layer (across all eight attention heads) are 0.275 for the self-edge and 0.185 for the neighbourhood edges.

---

### Public Comment · (anonymous) · 2017-11-21
**Computational Complexity and Experimental Results**

The proposed GAT needs to compute e_{i,j} for arbitrary i, j in the graph. Thus the total number of e_{i,j} is P(N,2) for directed graph and C(N,2) for undirected graph. When the graph size (N) increases, the computational complexity increases quickly. Can the author show the computational complexity and compare it with existing methods? Also, what is the definition of attentional mechanism "a" in eqn (1)?

In Kipf's GCN paper, the performance is claimed being the results of an average of 100 random initializations. However, this paper using the best performance to compare with others' average performance is not reasonable. From the last comment we are informed that the average performance for 10 runs of the same model with different random seeds are (with highlighted standard deviations): Cora: 83.0 +- 0.6 (with a maximum of 83.9%) Citeseer: 72.7 +- 0.7 (with a maximum of 74.2%). For a fair comparison with the baseline, the authors may provide the average performance for 100 runs.

Also, I am curious of the reason why the author did not show the Pubmed dataset results, which is used with Cora and Citeseer togethor in existing graph CNN works. Pubmed's graph size is much larger than the other two, so it is an important dataset to test the proposed method and compare with the baselines.

---

> ### Author Response · Authors · 2017-11-22
> **Further experimental results, and clarification of complexity**
>
> Thank you for your comments and queries on the complexity and experimental setup!
>
> We fully agree that 100-run performance would give the fairest comparison to the baselines. Accordingly, results after 100 runs of our model largely follow the trend of the 10-run result:
> Cora: 83.0 +- 0.7 (maximum 84.3%)
> Citeseer: 72.6 +- 0.8 (maximum 74.2%)
>
> To highlight: we have used a 1-run result (without any additional runs), rather than the best result, in the original writeup as submitted. Our N-run results already showed it is possible to achieve better single-run results than 83.3% and 74.0%, respectively.
>
> Our particular choice of attentional mechanism, a, is explicitly written out in Equation (3), clarified by the text immediately preceding this Equation, and illustrated by Figure 1 (left). It may be expressed as:
>
> a(x, y) = a^T[x||y]
> where a is a learnable weight vector, || is concatenation, and ^T is transposition.
>
> That is, it corresponds to a simple, linear, single-layer MLP with a single output neuron,  acting on the concatenated features of the two nodes to compute the attention coefficient---largely similar to the original attention mechanism of Bahdanau et al.
>
> Theoretically, our model needs to compute the attentional coefficients e_{i, j} only across the edges of the graph, i.e. O(|E|) computations of a single-layer MLP overall, which are independent, and thus can be parallelised. This is on par with other baseline techniques (such as GCNs or Chebyshev Nets). Taking into account that we need to perform a matrix multiplication on each node's features (to transform the feature space from F to F' features), we may express the overall computational complexity of a single attention head's computations as O(|V| x F x F' + |E| x F'), where F is the input feature count, and F' the output feature count---keeping in mind that many of these computations are trivially parallelisable on a GPU.
>
> The P(N, 2) or C(N, 2) values mentioned in your comment would correspond to a dense graph (E ~ V^2), where O(V^2) complexity is unavoidable regardless of which graph technique is selected.
>
> Unfortunately, even though the softmax computation on every node should be trivially parallelisable, we were unable to make advantage of our tensor manipulation framework to achieve this parallelisation, while retaining a favourable storage complexity (as its softmax function is optimised for same-sized vectors). This implied that we had to reuse the technique from the self-attention paper of Vaswani et al., wherein attention is computed over all pairs of nodes, with a bias value of -inf inserted into non-connected pairs of nodes. This required a storage complexity of O(V^2), and caused OOM errors on our GPU when the Pubmed dataset was provided---which is the reason for the lack of results on Pubmed.
>
> We were, however, able to run our model on the PPI dataset, which has 3x the number of nodes and 18x the number of edges of Pubmed. We were able to do this as PPI is split into 24 disjoint graphs (with test graphs entirely unseen), allowing us to effectively batch the softmax operation. This should still demonstrate evidence of the fact our model is capable of retaining competitive performance when scaling up to larger graphs (and, perhaps more critically, that it is capable of inductive, as well as transductive, generalisation).
>
> We thank you once again for your comments, and will be sure to include some  aspects of the above discussion in a revised version of the paper!

---

### Public Comment · ~Fabian_Jansen1 · 2017-11-24
**Spurious weights**

In equation 3 the coefficients are calculated as a softmax. However, it appears that the first half of the weight vector "a" beloning to the node "i" under consideration drops out of the equation and is thus not used nor trained.

From equation 3:
alpha(i,j) = exp(a * [W*h(i)||W*h(j)]) / Sum(k) exp(a * [W*h(i)||W*h(j)])

Writing vector a explicitly in two parts as a = [a(1)||a(2)]:

alpha(i,j) = exp([a(1)||a(2)] * [W*h(i)||W*h(j)]) / Sum(k) exp([a(1)||a(2)] * [W*h(i)||W*h(j)])
                 = exp(a(1)*W*h(i) + a(2)*W*h(j) ) / Sum(k) exp(a(1)*W*h(i) + a(2)*W*h(k) )
                 = exp(a(1)*W*h(i) ) exp(a(2)*W*h(j) ) / Sum(k) exp(a(1)*W*h(i)) exp(a(2)*W*h(k) )
                 = exp(a(2)*W*h(j) ) / Sum(k) exp(a(2)*W*h(k) )

The a(1) part drops out

---

> ### Author Response · Authors · 2017-12-19
> **Thank you!**
>
> Thank you very much for spotting this! We have now updated our method to make advantage of all the weights (by applying a simple nonlinearity to the output before normalisation), and will be sure to acknowledge you in the final version of our paper!

---

### Author Response · Authors · 2017-12-19
**Summary of revisions made to the paper in the discussion period**

We hope that the revisions we have made to the paper have properly addressed the comments of the reviewers as well as other researchers on our work - and that its overall contribution, quality and clarity is now significantly improved! We would like to thank everyone once again for their thoughtful comments on our paper.

We provide a summary of the changes made to the paper:

* We have been able to implement a sparse version of the GAT layer, allowing us to execute the model on the Pubmed benchmark. We make this clear across the document, wherever we enumerated the datasets under study.

* In Section 1, we have added appropriate references to relevant related work: MoNet, VAIN, neighbourhood attention, locally linear embedding (LLE) and memory networks.

* In response to Fabian Jansen’s comment (and as reiterated by one of the reviewers), we have now inserted a LeakyReLU nonlinearity to our attention mechanism---representing a minimal change from the previous mechanism’s properties, while no longer having spurious weights. Section 2.1 details this change (within Equation 3 and text immediately preceding it, Figure 1, and its caption).

* In Section 2.2, we no longer mention the storage limitation of our model, as we have been successful in addressing it (by implementing a sparse GAT layer). Instead, we mention the limitation of the current sparse matrix multiplication operation with respect to batching.

* In Section 2.2, we incorporate many of the useful comments (from reviewers and other researchers) about the characteristics of our model: time complexity (especially, comparing it to our primary spectral baselines), the effects of multi-head attention on the parameter count, clarifying the computational/performance tradeoffs compared to GraphSAGE, detailing the relationship between GAT and MoNet, an informal assessment of the suitability of GPUs for such computations, and comments about the model’s effective “receptive field” size around each node and the computational redundancies of the model.

* In Section 3.2., we state that we now compare our model with the results reported by the MoNet paper as well.

* In Section 3.3., we corrected two typos (dropout p = 0.6, rather than 0.5; also, our early stopping strategy took into account both the loss and the accuracy), and noted the slight differences in architecture we used for the Pubmed dataset. We also make explicit the inductive learning experiment of a GAT model with attention turned off (as recommended by one of the reviewers).

* In Section 3.4., we now report the new results of all models considered, after 100 runs for transductive tasks (for fairly comparing against the work of Kipf and Welling), and 10 runs for inductive tasks. We also provide the best 100-run transductive results we were able to obtain with a GCN model computing 64 features (with ReLU or ELU activation), and the best inductive result we were able to obtain with GraphSAGE (by only changing its architecture, and not the sampling strategy), as well as the 10-run result of the aforementioned inductive GAT model with attention turned off (as a comparison to a GCN-like model computing the same number of features). These results are now all enumerated in Tables 2 and 3, and are discussed appropriately in the main text body. The tables’ captions have been expanded to make the result presentation more clear as well.

---

### Decision · Program_Chairs · 2018-01-29
**ICLR 2018 Conference Acceptance Decision**

**Decision:**

Accept (Poster)

**Comment:**

The authors appear to have largely addressed the concerns of the reviewers and commenters regarding related work and experiments. The results are strong, and this will likely be a useful contribution for the graph neural network literature.